# Reducing the Need for Backpropagation and Discovering Better Optima With Explicit Optimizations of Neural Networks

## Abstract

Iterative differential approximation methods that rely upon backpropagation have enabled the optimization of neural networks; however, at present, they remain computationally expensive, especially when training models at scale. In this paper, we propose a computationally efficient alternative for optimizing neural networks that can both reduce the costs of scaling neural networks and provide high-efficiency optimizations for low-resource applications. We derive an explicit solution to a simple feed-forward language model (LM) by mathematically analyzing its gradients. This solution generalizes from singlel-layer LMs to the class of *all* single-layer feed-forward softmax-activated neural models trained on positive-valued features, as is demonstrated by our extension of this solution application to MNIST digit classification. For both LM and digit classifiers, we find computationally that explicit solutions perform near-optimality in experiments showing that 1) iterative optimization only marginally improves the explicit solution parameters and 2) randomly initialized parameters iteratively optimize towards the explicit solution. We also preliminarily apply the explicit solution *locally by layer* in multi-layer networks and discuss how the solution's computational savings increase with model complexity—for both single- and mult-layer applications of the explicit solution, we emphasize that the optima achieved *cannot* be reached by backpropagation alone, i.e., *better* optima appear discoverable *only after* explicit solutions are applied. Finally, we discuss the solution's computational savings alongside its impact on model interpretability and suggest future directions for the derivation of explicit solutions to complex- and multi-layer architectures.

## 1 Introduction

The burgeoning field of artificial intelligence (AI) has seen a rapid expansion in the development and training of increasingly large-scale models in the era of deep learning. With their escalating size and complexity, these models have exhibited substantial improvements in performance, attracting considerable research interest and investment. Among these models, two primary categories stand out: large language models (LLMs), including GPT-3 (Brown et al., 2020), and LLaMA (Touvron et al., 2023) (to name a few); and diffusion probabilistic models (Ho et al., 2020), such as DAll-E (Ramesh et al., 2021) and Stable Diffusion (Rombach et al., 2022). Both paradigms demonstrate superlative capabilities in their respective domains of natural language processing (NLP) and computer vision. The potential of these models is further amplified by combining text, images and other modalities to construct even more powerful models, exemplified by the likes of KOSMOS-1 (Huang et al., 2023) and GPT-4 (OpenAI, 2023).

Despite making impressive strides in AI, our collective understanding of the inner workings of these models is far from complete. The absence of a comprehensive understanding of their internal mechanisms impedes our ability to fully exploit their capabilities while simultaneously raising various challenges (Bommasani et al., 2022). One of the primary concerns is the reliability and safety of these models. LLMs are prone to generating biased and unreliable text, while diffusion models may produce distorted images that conflict with basic human perception. The unpredictable behaviors of these models in novel or unusual situations challenges their operational benefits to humans via their (in)abilities to avoid inadvertent harms (Kenton et al., 2021; Weidinger et al., 2021)). Concurrently,

efficiency is another major concern (Shen et al., 2023). Backpropagation is their predominant training and optimizing method, and entails a high computational cost, particularly as models scale up their iterative gradient computations for optimization ((Rumelhart et al., 1986a), (Rumelhart et al., 1986b)). Training such large models requires a substantial amount of data, also necessitating significant efforts in data processing.

In light of these challenges, how can we ensure that these models are reliable, interpretable, and efficient? We posit that understanding the optimization processes underlying these models is crucial. Perhaps, grasping the intricacies of model optimization will allow a more straightforward approach, requiring fewer iterations to achieve the same or better quality results? Furthermore, understanding how models optimize allows us to adjust specific parameters in the weight matrices, enabling models to perform in a desired manner. In this paper, we investigate by starting from a simple form of neural network: the single-layer feed-forward neural network. We propose an "explicit" optimization of it, and argue that explicit optimization offers a vital alternative to the current trends in neural network training. By providing this computationally efficient and interpretable method of optimization, this approach has the potential to significantly accelerate progress in the field of AI.

To substantiate our proposed solution to parameter optimization, we conduct computational experiments on language modeling and digit classification to demonstrate its near-optimal performance. Our findings indicate that iterative optimization (requiring backpropagation) only marginally improves upon the parameters of the explicit solution, and that randomly initialized parameters gradually converge towards the explicit solution through iterative optimization. By following a similar experiment to our language modeling tests, we also conduct preliminary evaluations on the MNIST (LeCun & Burges, 1998) dataset. These results demonstrate the encouraging usage of explicit optimization as a strategy for improving the efficiency of training neural networks, in general, and at enhancing the interpretability of models. Furthermore, our findings underscore the potential of explicit optimization as a viable and efficient method for more complex, multi-layer models.

## 2 Deriving explicit optimization for a single-layer LM

This derivation originally began by analyzing word2vec's continuous bag-of-words (CBOW) variant (Mikolov et al., 2013a;b). Following this, we not only generalized the analysis to simple single-layer LMs, but ultimately, to all feed-forward neural networks with arbitrary non-negative feature sets, as it is now presented in **Appendix A**.

To define a single-layer model's explicit solution, one must define the data of prediction. We assume sequential data: a model's objective is to reconstruct a matrix $Y \in \{0,1\}^{M \times N}$ of unit-normalized rows: $\|Y_{m,:}\|_1 = 1$, which correspond to the set of target elements for prediction in the sequence. The sequence of predictions will be based on $M$ sets of matrix-features, contained in a tensor storing $K$ numerical vector-features of dimension $D$ for each $m = 1, \cdots, M$: $\mathbf{X} \in \mathbb{R}^{M \times K \times D}$. In other words, each $m$-target $Y_{m,:}$ has a corresponding slice from $\mathbf{X}_{m,:,:} \in \mathbb{R}^{K \times D}$ that is a matrix of $K$ vector features, drawn from other rows of $Y$. Specifically, each $m$-prediction has each $k = 1, \cdots K$ of its features drawn from the previous $K$ rows in $Y$ before the $m^{\text{th}}$: $\mathbf{X}_{m,k,:} = Y_{i_{m-k},:}$, defining the auto-regressive nature of the LM. Here, $i \in \{1, \cdots, N\}^M$ is the vector of target indices for each prediction in the sequence of $M$, and tokens early in documents—with $m < K$—are padded with `"<pad>"`.

The derived statement of optimization is defined for a *decoder*-matrix, $U \in \mathbb{R}^{D \times N}$ under the action of the softmax function: $\varphi$, defined as: $\hat{Y}_{m,:} = \varphi(H_{m,:}U)$, where hidden states for each prediction are computed as the sum of vector-features: $H_{m,:} = \sum_{k=1}^{K} \mathbf{X}_{m,k,:}$. Under this notation, the LM is optimized by the cross entropy loss, or, *negative* log-*likelihood* of model prediction probabilities:

$$L = -\sum_{m=1}^{M} \log \varphi(H_{m,:}U)_{i_m} \tag{1}$$

### 2.1 An explicit solution for single-layer optimization

To understand the explicit solution's derivation and potential applications, it is helpful to state:

**Definition**: A data set of vector-inputs $\boldsymbol{H} \in \mathbb{R}^{M \times D}$ and -outputs $\boldsymbol{Y} \in \mathbb{R}^{M \times N}$ has generalized co-occurrences $\boldsymbol{F}(\boldsymbol{H}, \boldsymbol{Y}) \in \mathbb{R}^{D \times N}$ between inputs and outputs defined by the sum of outer products:

$$\boldsymbol{F}(\boldsymbol{H}, \boldsymbol{Y}) = \sum_{m=1}^{M} \boldsymbol{H}_{m,:} \otimes \boldsymbol{Y}_{m,:} = \boldsymbol{H}^T \boldsymbol{Y}. \tag{2}$$

In **Appendix. A**, we go on to show that this definition is critical to softmax-activated optimization:

**Theorem**: A softmax-activated feed-forward layer receiving $K$-norm non-negative $D$-dimensional inputs $\boldsymbol{H}_{m,:}$ for each target of prediction $\boldsymbol{Y}_{m,:}$ is approximately optimized by a column-wise translation of the layer's generalized log-co-occurrence matrix: $\boldsymbol{U}_{j,i} = \log \boldsymbol{F}(\boldsymbol{H}, \boldsymbol{Y})_{j,i} + w_i$. The translating weights, $w_i$, are defined by $i$-column (output) as: $w_i = \frac{K-1}{K} \log(\sum_{d=1}^{D} \boldsymbol{F}(\boldsymbol{H}, \boldsymbol{Y})_{d,i})$, defining an explicit form for each of the layer's $j, i$-parameters by the expression:

$$\boldsymbol{U}_{j,i} = \log \boldsymbol{F}(\boldsymbol{H}, \boldsymbol{Y})_{j,i} - \frac{K-1}{K} \log \left( \sum_{d=1}^{D} \boldsymbol{F}(\boldsymbol{H}, \boldsymbol{Y})_{d,i} \right) \tag{3}$$

Proof of the above is provided in **Appendix A** for reference. In general, we'll refer to $K$ as a *priming number*. In circumstances where features are not unit-normalized (but still positive) the explicit solution *also* appears to function quite well. However, one must extend the priming number from the discrete number of features to an estimate of this as the average norm of a given feature vector: $\hat{K} = (\sum_{m=1}^{M} \sum_{d=1}^{D} \boldsymbol{H}_{m,d})/M$. However, it turns out that the most critical knowledge required for the explicit solution's use in application is *understanding explicit forms for given softmax classifier's inputs and outputs*. For a decoder—such as $\boldsymbol{U}$ in the theorem—it is often quite clear what the inputs (features) and outputs (supervising targets) are. While the explicit solution may be applied *locally*, to individual layers of, multi-layer networks, *compositional* optimization (not covered in this work) of multi-layer and attention-supported networks require further investigation. Regardless, one assumption for this theorem should be empirically considered in computation alongside prediction experiments using the explicit solution: the degree to which token counts scale with their average of geometric means of co-occurrences.

## 2.2 Evaluating the Mean Co-occurrence Scaling Assumption over Data

Throughout this work, experiments will be focused on utilizing single-layer language models that generalize as feed-forward neural networks, and the data that will be used for these experiments is provided by the BabyLM Challenge (Warstadt et al., 2023; Gao et al., 2021), which is a shared language modeling task that "challenges community members to train a language model from scratch on the same amount of linguistic data available to a child." We see this as both an excellent opportunity to engage a current interest in language modeling, as well as demonstrate if explicit optimization can produce more effective learning over little data. From this data set, we utilize the smallest training set of 10-million tokens, which is quite small from a language modeling perspective.

Before considering the performance of the feed-forward model's explicit optimization, we first test its assumption over the training data. This is done for radius: $K = 1$, and should be studied more closely for other values in future work. As can be seen for $K = 1$, a scaling relationship is visibly present in **Fig. 1**; however, since the slope observed likely doesn't *exactly* fall along the line $y = x$ (slope 1), a better set of weights could likely be taken over the proven case, i.e., instead it may be that $\boldsymbol{w}_i \propto \boldsymbol{f}_i^{(1-K-\delta)/K}$ is optimal, for some small, positive value of $\delta$. Furthermore, since the exact computation of $\boldsymbol{w}_i$—as the values $\langle \mathbb{E}_G [\boldsymbol{F}_{j,i} \mid \boldsymbol{H}_{m,:}] \rangle$—requires essentially the same computational cost as running a single training epoch of gradient-based optimization, one might ask: how much additional optimization can be done on top of the solution provided by **Eq. 5**, provided iterative optimization is used?

## 3 Computational Experiments

The choice to utilize data from the BabyLM challenge for language modeling is twofold: as a benchmark, BabyLM provides comparative information against systems submitting to the challenge, while its relatively small size allows for rapid prototype iteration. Interested in the experimental

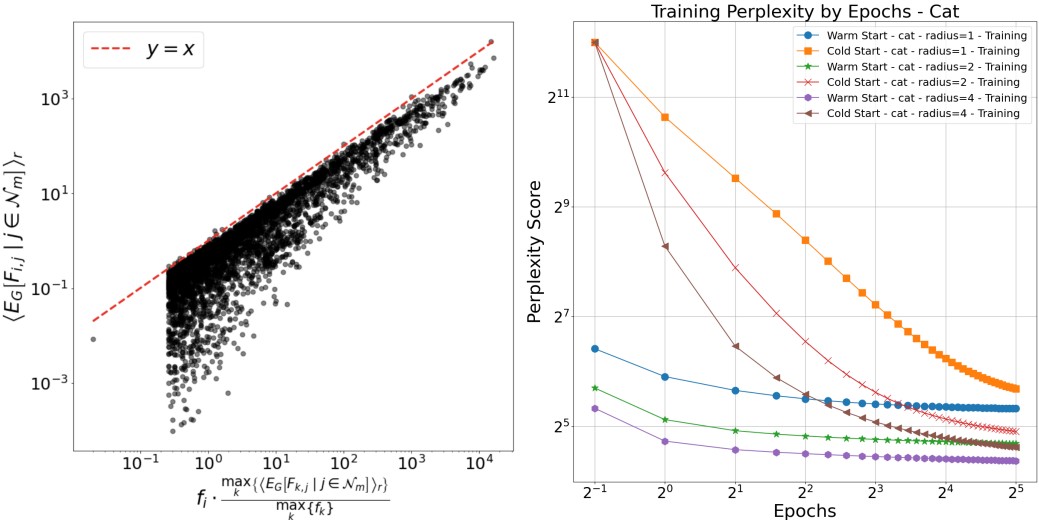

Figure 1: (Left) The **Cat** and **Sum** models are equivalent when $K = 1$ (depicted), and for this case the scatter plot shows the scaling relationship between unigram frequency: $\boldsymbol{f}_i = \sum_{j=1}^{N} \boldsymbol{F}(\boldsymbol{H}, \boldsymbol{Y})_{i,j}$ and the power mean of geometric averages of in-neighborhood co-occurrences, spanning the entire training data set. (Right) Training set loss curves from iterative optimization in the case of cold (random parameters) and warm (explicit solution parameters) start for the **Cat** model. Axes depict the perplexity as a function of epoch (iteration) on a logarithmic scale for a total of 32 epochs. Note: model loss *prior* to training is likewise reported in the loss curves, with values presented at epoch $0.5$ for visual clarity, as a means of observing the degree of optimally achieved by the explicit solution as a starting point. Additional loss curves are for the development set, as well as for both sets and the **Sum** model in Supplementary Material **Sec. B**.

benefits offered by small size, we conduct all BabyLM experiments on its smallest training set of 10-million tokens, and then sample down to $10\%$ of its development data to monitor the possibility of early stopping. All language modeling experiments use Adagrad (Duchi et al., 2011) and a learning rate of $\eta = 0.01$ to optimize models for 32 epochs (iterations) and use no dimensionality reduction, which is left for future work (discussed in **Sec. 5**). However, this requires the use of a vocabulary-sized embedding layer, which presents a hard constraint on the feasible vocabulary sizes that may be used in system memory. Thus, we follow (Sennrich et al., 2016) and utilize the byte-pair encoding algorithm for tokenization, limiting the number of merge rules to fix a vocabulary size of $2^{12} + 2$ tokens, reserving the +2/additional slots used for an out-of-vocabulary token and a padding token, the latter of which assures all points of prediction are based on exactly $r$ features. Finally, as a demonstration of the feed-forward optimization's generalization over data domains, this section's final sub-section will describe preliminary computational experiments on MNIST (LeCun & Burges, 1998), demonstrating the explicit optimization's performance on handwritten digit classification.

### 3.1 LANGUAGE MODELING EXPERIMENTS

The architecture analyzed in **Sec. 2** is augmented in two ways. First, since co-occurrence matrices are sparse, LM-experiments require a model of noise to fill co-occurrence zeros with small positive values. Second, alongside sum-based aggregation, we explore the improvements obtained from radial-concatenation of inputs, which increases parameter complexity and performance, while demonstrating the optimization's capacity for architectural generalization.

### 3.1.1 MODELING NOISE IN OBSERVATIONS

To assure that generalized co-occurrence matrices are dense, we likewise densify vectors using a model of noise. This is done by first computing a vector of token counts $\boldsymbol{f} = \sum_{m=1}^{M} \boldsymbol{Y}_{m,:}$, and from it, the average (un-noised) basis vector: $\overline{\boldsymbol{e}} = (\sum_{n=1}^{N} \boldsymbol{f}_n e^{(n)})/M$, as well as a model: $\boldsymbol{q} \in (0,1)^N$,

for the portion of occurrences that each $n$-token's observations are (non-)erroneous. Assuming that the highest-count tokens will be the least erroneously observed, we assume that only one error will be observed relative to each token's observed count, that is: $\boldsymbol{q}_n = \boldsymbol{f}_n/(\boldsymbol{f}_n + 1)$. Next and regardless of the token that is observed, we wish to modify its standard basis vector according to the probabilities that any different, $j$-token, should have been observed, instead, which will take the form of a normalized ($\|\boldsymbol{p}\|_1 = 1$) *noise* vector: $\boldsymbol{p} \in (0, 1)^N$, defined to be near-uniform as: $\boldsymbol{p} = (1 - \overline{\boldsymbol{e}})/\|1 - \overline{\boldsymbol{e}}\|_1$. To understand $\boldsymbol{p}$ intuitively, we first note that 1-minus each of the average embedding $\overline{\boldsymbol{v}}$'s (normalized) value is also a probability, which expresses the chance that a given dimension's magnitude is spurious (should not be observed). In application, the value of each *embedding* in the matrix $\boldsymbol{E} \in \mathbb{R}^{N \times N}$, is finalized by adding noise to rows in an embedding layer:

$$\boldsymbol{E}_{n,:} = \boldsymbol{q}_n \boldsymbol{E}_{n,:} + (1 - \boldsymbol{q}_n)\boldsymbol{p} \tag{4}$$

which is utilized to re-define the inputs-tensor densely: $\mathbf{X}_{m,k,:} = \boldsymbol{E}_{\boldsymbol{i}_{m-k}}$ for each $k = 1, \cdots, K$ of token $m$'s input feature-vectors.

### 3.1.2 Context models and generalized co-occurrences

To define features for each $m^{\text{th}}$ token according to its $K$ previous, noisy/dense embedding-vectors, we consider two conditions (architectural variants). The first considers the sum of vectors from $\boldsymbol{E}$—mapped into $\mathbf{X}$—from the $K$ previous tokens within the radius of $K$: $\boldsymbol{H}_{m,:} = \sum_{k=1}^{K} \mathbf{X}_{m,k,:}$. We note once again that tokens early in documents—with $m < K$—are padded with the `"<pad>"` token. While aggregations defined by sums are indicated by **Sum** in tables, figures, and experiments; the second variant considered featurizes each $m^{\text{th}}$ token according to $\boldsymbol{E}$-vector *concatenation*, and so is indicated by **Cat** in tables, figures, and experiments. Hidden states for **Cat** models are ultimately expressed quite similarly: $\boldsymbol{H}_{m,:}^{(\text{Cat})} = [\mathbf{X}_{m,k,:}]_{k=1}^{K}$. Regardless of featurization, a dense co-occurrence matrix is now concisely expressed by the sum of outer products of standard-basis output vectors with their dense input vectors, i.e., we compute the matrix $F(\boldsymbol{H}, \boldsymbol{Y})$ for **Sum**-explicit solutions, and *instead* compute $F(\boldsymbol{H}^{(\text{Cat})}, \boldsymbol{Y})$ for the **Cat**-model solutions.

### 3.2 MNIST handwritten digit classification

For digit classification, MNIST requires determining a label from $\{0, 1, 2, 3, 4, 5, 6, 7, 8, 9\}$ for a data set of $M = 60,000$ (non-sequentially ordered) images. Thus, we consider each $m$ of MNIST's training instances as having targets within a matrix $\boldsymbol{Y} \in \{0, 1\}^{M \times 10}$, whose rows are unit-normalized: $\|\boldsymbol{Y}_{m,:}\|_1 = 1$. The input images, themselves, are contained in a $28 \times 28$-slice of an input matrix $\mathbf{X} \in \mathbb{R}^{M \times 28 \times 28}$. MNIST images are numerically structured as $28 \times 28$ matrices of integer values that range over $0, \cdots, 255$. To pre-process MNIST data, we translate all values by (add) 1 to ensure positivity, as well as normalize (divide) each integer by 256 to assure all input-values fall within the range $(0, 1]$; these are then flattened to define feature vectors in the matrix $\boldsymbol{H} \in (0, 1]^{M \times 784}$ according to the equation: $\boldsymbol{H}_{m,:} = \text{flatten}((\mathbf{X}_{m,:,:} + 1)/256)$. The standard train-test split is then processed, using the derived optimization presented in **Sec. 2**. Much as we do for LM'ing experiments, we likewise present the results of backpropagation-based iterative optimization on MNIST models, following our application of their explicit solutions. This includes our preliminary investigation into multi-layer models, which uses the derived explicit solution locally by layer (see **Sec. 3.3**). However, to proceed with any of these experiments more consideration is needed regarding MNIST's pre-processing, e.g., since $\boldsymbol{H}_{m,:}$-input vectors don't all have the same norm.

### 3.2.1 MNIST model priming

Without an available analytical defintion of $K$ for the MNIST model's architecture, we determined to scan priming numbers and compare model accuracy. We generally hypothesize that the MNIST model's priming number is related to the norms of input vectors: $\boldsymbol{H}_{m,:}$. This was conducted over the integer range of $k \in \{1, \cdots, 784\}$, whose upper limit is the flattened dimensionality—and maximum norm of—of a pre-processed MNIST training instance (784). With these values of $k$, the priming numbers our scan covered defined normalization weights as: $k \mapsto \boldsymbol{f}_i^{-(k-1)/k}$, which generalized our interpret of the single-layer LM's normalization being a function of the *radius*: $K \mapsto \boldsymbol{f}_i^{-(K-1)/K}$, which in that scenario is both the number of features per prediction *and* the 1-norm of each given feature vector.

Table 1: Training (T) and development (D) set perplexities from language modeling experiments for the **Sum** and **Cat** models. Suffix -p indicates the performance of pretrained models optimized by the explicit solution prior to iterative optimization; suffix -c indicates cold-start models with randomly initialized parameter matrices; suffix -w indicates warm-start models that use the explicit solution as a starting point for iterative optimization. For -c and -w, perplexity is reported following 32 epochs of optimization with the Adagrad optimizer and a learning rate of $\eta = 0.01$. Since **Sum** and **Cat** are equivalent when $r = 1$, results are copied (not reran) between the two below to support trend interpretation.

| $K$ | Sum | | | | | | Cat | | | | | |
|---|---|---|---|---|---|---|---|---|---|---|---|---|
| | T-c | D-c | T-p | D-p | T-w | D-w | T-c | D-c | T-p | D-p | T-w | D-w |
| 1 | 51.4 | 56.4 | 85.4 | 91.8 | 40.1 | 45.7 | 51.4 | 56.4 | 85.4 | 91.8 | 40.1 | 45.7 |
| 2 | 47.6 | 56.3 | 104.0 | 121.5 | 41.0 | 51.3 | 30.0 | 35.6 | 52.0 | 61.8 | 25.8 | 33.0 |
| 4 | 60.7 | 75.0 | 144.0 | 176.9 | 53.8 | 72.2 | 24.6 | 31.6 | 40.3 | 55.4 | 20.7 | 31.2 |

## 3.3 LOCAL APPLICATIONS OF THE EXPLICIT SOLUTION TO MULTI-LAYER NETWORKS

To extend the explicit solution to multi-layer networks we take a naïve first approach. This is for both simplicity and the method's potential utility, leaving detailed investigations into multi-layer warm-starts and compositional optimization for future work. The explicit solution is specifically known for softmax-based activation, and thus, one shouldn't expect to find explicit solution values perform well when an activation functions are changed. Likewise, since this work does not consider dimensionality reduction, our current options are limited to the input and output dimensionalities, which can't create useful bottlenecks, which are likely need for truly enhanced multi-layer prediction. Nevertheless, one can naïvely 1) train a single-layer classifier, $U$, and then 2) define *second-order* hidden states: $H^{(2)} \in \mathbb{R}^{M \times N}$ by first-layer outputs: $H^{(2)}_{m,:} = \varphi(H_{m,:}U)$, and 3) subsequently define a second layer's decoder by applying the explicit solution over the co-occurrence matrix: $F(H^{(2)}, Y) \mapsto U^{(2)} \in \mathbb{R}^{N \times N}$. This procedure is used to train all 2-layer warm-start models, which are applied to MNIST experiments (only). While we consider if this *local optimization* strategy using the explicit solution could be fully developed into a generalized multi-layer warm start, we ultimately leave these questions to future work, since full functionality in multi-layer warm-starts requires dimensionality reduction and more understanding in activation function processes.

## 4 EXPERIMENTAL RESULTS

In all MNIST experiment pre-processing, images were flattened, added to 1, and divided by 256, which results in the training set's average norm being approximately 105.99. We can immediately see from the priming number scan (at right in **Fig. 4**) that the initial $\arg\max$ value for a priming-number *is* the average norm. Using the average norm, the explicit solution demonstrates non-trivial MNIST classification, achieving 82.86% test set accuracy. At left in **Fig. 4**, the single-layer MNIST cold-start required roughly 5 epochs of learning to reach the explicit solution's starting accuracy of roughly 82%, while the 2-layer (see **Sec. 3.3**) cold-start suffered from parameter disorientation that ultimately left it performing poorly. Warm-starts, however, begin iterative optimization at 2–4-times higher accuracy, and terminates early according to a naïve early-stopping criterion (increased value of cross-entropy loss) sooner. This property is observed for both 1- and 2-layer models, i.e., demonstrating no loss of performance for including a second layer when the warm-start is used. While the single-layer warm-start's early stopping resulted in roughly 4-times less computation, requiring only 23 vs. 93 epochs of training for optimization, the two-layer cold-start never stopped. In both 1- and 2-layer cases, warm-starts led to the highest accuracies: 92.57% for the 1-layer warm-start, and 92.93% for the 2-layer warm start. These values ultimately compared to 91.68% for the 1-layer and 75.64% for the 2-layer cold-start.

Language modeling experiments are discussed in terms of average perplexity: $e^L/M$ for both of the **Sum** and **Cat** model variants, which again, are equivalent when $K = 1$. Their hyperparametric articulations are recorded both in **Tab. 1** and in **Fig. 1**; this includes the scenario of models that are pre-trained by the explicit solution prior to any iterative optimization (suffix -p in **Tab. 1**); cold-start models, whose parameter matrices are randomly initialized (suffix -c); and warm-start models that utilize the explicit solution as a starting point for subsequent iterative optimization (suffix -w). At a

high-level, we note that **Cat**-based models generally achieve lower perplexity (better performance) than their **Sum**-based counterparts, and that explicit solutions initialized the best models, i.e., that models could be improved beyond the point of explicit solution initialization.

Deriving the $w$-weights that optimize the FF-model's solution required assuming that a scaling relationship exists between an average of co-occurrence averages and token counts. The empirical reality of this apparent-but-noisy relationship is depicted by the cone-shaped linearity in the left panel of **Fig. 1** for both/either of the **Cat** and **Sum** models, which are equivalent when $K = 1$. When this same figure was depicted for larger-radius sum values (e.g., $K = 2$ and $4$) in early testing, the scaling relationship was noted diverge somewhat more; this notably appeared to correspond with decreases in performance (increases in model perplexity) observed at larger radii in **Tab. 1**, where perplexity values are stored. However, we note that when poorer-quality scaling relationships of the kind depicted in **Fig. 1** are observed (for larger radii), it doesn't appear to invalidate the explicit solution gravely, that is, the explicit solution still greatly improves performance over a cold-start. Nevertheless, ramping up to full runs on BabyLM's 10-million token training sample during this work's development made it clear that the scaling relationship likely *stabilizes* with bigger data. This is a subject that could be interesting to follow up on in future studies. Finally, while the scaling relationship appears to stabilize as data is increased, **Fig. 1** also guides our future, more refined analysis into the scaling relationship, as the slope of the scaling relationship is likely a small amount shallower than the line $y = x$ in **Fig. 1**. Put differently, refining the analysis of this relationship to better reflect the empirical scaling relationship would likely lead to an improved explicit solution, which could be a promising avenue for future study.

In **Tab. 1**, training and development set losses can clearly be seen to co-optimize, with larger percent gaps between training and development loss appearing for larger context windows indicating insufficient generalization, and perhaps the need for more training data when more features are used (larger $r$-radius). Moreover, related effects can be observed when comparing the **Sum** and **Cat** models with increasing radius. Almost paradoxically, perplexity *increases* for the **Sum** model at larger radii, while it progressively *decreases* for the **Cat** model as larger radii (more features) are utilized. The pattern of loss-increase with radius can be seen throughout nearly all of the **Sum** experiments— except the cold-start at $k = 2$—demonstrating the likely need for attention-upon-aggregation when vectors are summed across position, in-line with the transformer-architectural developments, presenting an architectural avenue of study for future work in reducing **Sum**-based architecture perplexities through integration and analysis of attentive mechanisms. Furthermore, we note that the cold-start model performing 'best' at $k = 2$ is likely a sign of less-stable optimization, particularly, since cold-starts are randomly initialized. This remark is parallel to our observation throughout testing that explicit solutions demonstrate monotonic performance scaling, i.e., perplexity exclusively increases or decreases as hyperparameters are increased. This observations is an indication of a strong *ablation advantage* provided by the explicit solution—not only for its efficiency at producing comparable models and evaluations, but moreover, for the *deterministic* consistency that its provides. Juxtapose this to the erratic ablataion properties exhibited by the randomly-initialized models, i.e, whose **Sum** models have *lowest* at a bounded value: $K = 2$. This likely spurious result may have emerged from one 'lucky' initialization in the table's 6 cold-start models. Comparatively, our experimentation with the explicit solution leading up to the **Sum**-**Cat** experiments left zero doubt by demonstrating so much consistency **Sum** models performing worse with increased radius. Without the explicit solution, this could well have misguided our investigation, e.g., away from considering **Cat**-based models, and would have left us only with the option to 'waste' computation on redundant experiments averaging performance from different random initializations at much larger expense.

Models on the **Cat**-side of **Tab. 1**'s experiments more-intuitively drop in perplexity as as the radius is increased, and demonstrate the diminishing returns gained from increased positional information, i.e., indicating that most predictive information is present close to the point of prediction. The smooth reduction in perplexity—juxtaposed to *increases* seen for **Sum** models—indicates the utility of the outer product as a lossless (sparse) featurization pre-processor for the integration of independently-measured features; that is, since the concatenation of input vectors interacts with the outer product of token type *and* position, independently (this is also why **Cat** models have $K$-times the parameters of their **Sum** counterparts). In the right panel of **Fig. 1**, training loss by epoch is presented for **Cat** models (see Supplementary Materials **Sec. B** for additional loss curves). Here, the impact of a warm start can be seen on a logarithmic scale, where over the 32 iterations provided for optimization, cold-start models ended not far from where warm-starts began. For small exper-

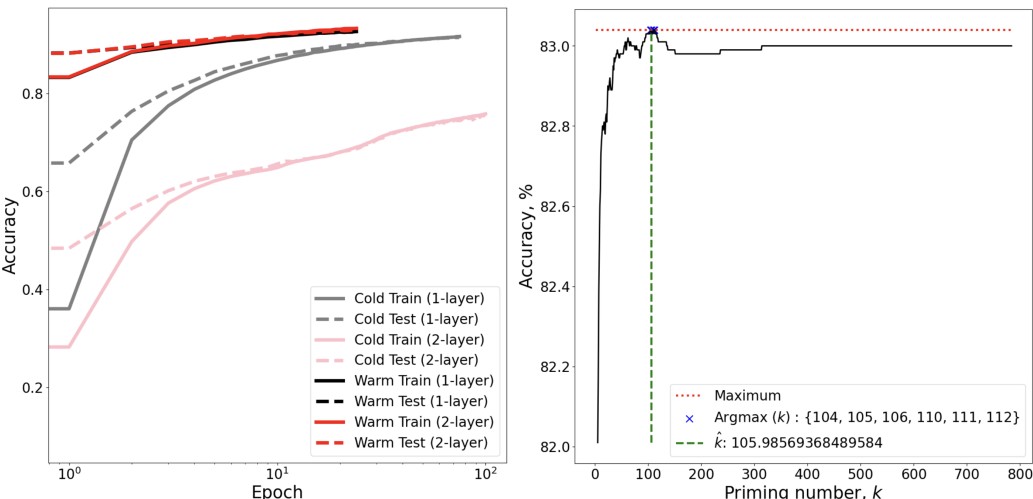

Figure 2: (left) Cold- and warm-start models for the MNIST data set and task for 1- and 2-layer models. (right) Values of the priming number are scanned for a single-layer MNIST model and compared on the basis of the explicit solution's accuracy.

iments like these, the efficiency benefit is essentially a 50% reduction of training cost; however, for larger/more-parametrically complex models trained over larger quantities of data, this reduction may well magnify.

## 5  DISCUSSION

Our optimization of models that have been warmed-up by the explicit solution is equivalent to the general case of 1) assuming (freezing) a matrix model (of co-occurrences) and then 2) determining a best intercept for a feed-forward architecture. This separate-intercept implementation was actually the intial (equivalent) formulation used for MNIST experiments, and both implementations are provided as software with this paper's release. The relative ease of using the MNIST data set to train a predictive model on an entirely different—numerical vectors vs, categorical text—data domain underscores the potential breadth of application of which the explicit solution is capable. Thus, an additional reason for performing MNIST's experiments (and providing their software) using a different implementation is to present the explicit solution in the context of a general featurizer, which could potentially be modified, e.g., to utilize a convolutional processing in the future. In fact, the sliding window technically does this for language modeling over one dimension, using a padded and uniform 1-dimensional convolution. We likewise present the explicit solution as a local optimizer, demonstrating have warm starts can naïvely be applied to individual layers of parameters, from the bottom of a given network. Thus, to produce more complex deep network architectures—as was proposed in (Hinton, 2022)—one can apply the explicit solution, from the bottom up first, to optimize a network non-iteratively/locally, and then subsequently—*compositionally*—apply backpropagation. However, *unlike* (Hinton, 2022), our approach *does not* change the downstream objective (it is still a softmax), allowing subsequent iterative optimization to be conducted to great effect, on top of warm starts. This multi-layer process is *efficient*, since the explicit solution's optimization requires information transmission in only the forward direction; that is, it requires only a single pass over the data for an accumulation of potentiation by its vector-statistics, even when applied to multi-layer networks.

A chief limitation of this work is its use of high-dimensional embedding vectors (in $\boldsymbol{E}$). While these vectors, which are similar to standard basis vectors, provide a degree of transparency, an effective dimensionality reduction strategy is critical. Ideally, since this approach advances local/independent optimization for multi-layer networks, a dimensionality reduction procedure that reduces from the standard-basis dimensionality must explicitly be predetermined and separate from the optimization process. It is possible to assume pretrained vectors from other algorithms for a fixed vocabulary; however, this assumption is challenged by the explicit solution's need for positive-valued features.

In particular, low-dimensional token vectors obtained from GloVe, GPT-2, etc. will produce components with negative values. Co-occurrences with the target one-hots and these negative values will thus result in partially negative-valued co-occurrence matrices, and the explicit solution should be proportional to the *logarithm* of these matrices. While the domain of the logarithm can be extended to negative numbers via the complex number system, this would result in the need to formalize softmax prediction as a function of the *complex* domain. Assuming $\boldsymbol{U}$ and $\boldsymbol{H}$ are taken over the complex field $\mathbb{C}$, one need only compute $\varphi\varphi^*(\boldsymbol{H}_{m,:}\boldsymbol{U})$, too, since $\varphi\varphi^*$, or, the vector-output of *magnitudes* of the values returned by the softmax function (complex conjugate products) is a probability distribution. This would undeniably present an underlying quantum nature—and perhaps expressive benefits to prediction—which could well be considered in future extensions of this work.

Another limitation of this work is its *local* focus on applying the single-layer solution to multi-layer models. While a local-optimization strategy for multi-layer networks works, it doesn't allow for useful transition to more efficient hidden activation functions, and moreover, compositional optimizations of multilayer models have substantial performance benefits that should not be left unconsidered by mathematical analysis, i.e., there's no reason to expect this work can be extended to compositional explicit solutions for multi-layer structures. Similar analyses could be framed for multi-layer models, but it must be noted that, *locally*, optimization takes the same form (i.e., the proof generalizes) of $\log$-conditional probability with respect to the given the inputs and outputs of a softmax function. This is why (mathematically) it should be possible to chain local optimizations of the derived solution for a first-order approximation of complex-model parameters. However, it also means it would be possible to directly apply the explicit solution to attention distribution, *if* one knew the 'hidden' targets of an attention distribution, i.e., answering "how should features be weighted to improve performance?" Again, the local-optimizing approach *does not* capture compositional differential information, and so will always need to be followed by backpropagation to achieve performant models. For fully compositional differential optimization, explicit solutions can be set up as solutions to more complex sets of differential equations, and if not explicitly solved, at least approached with the various high-performance differential equation approximators that exist. These avenues can be considered in future work, and movement in this direction is critical for the efficient optimization of neural architectures.

Alongside the generalization of the priming number to MNIST's numerical feature-input, it is perhaps the most surprising result from the language modeling experiments that the performance of the **Sum** model of input-vector aggregation worsened as features were added (aggregated); this result contrasts with the (expected) performance improvement realized with added features (radius size) in the **Cat** model. This effect possibly occurs because the *uniform* aggregation in the **Sum** fails to adequately weight the $K$ features. This is precisely the 'attention' problem that culminated in the use of self-attention, and the subsequent predominance of the transformer architecture. However, this result is not simply an interesting observation but also an *opportunity* for derivations of explicit, attention-based solutions to demonstrate performance improvements. Such a reduction in perplexity with larger $K$ values in a **Sum** model constitutes a 'smoking gun' for architects interested in extending this analysis to an explicit optimization of transformer-like architectures; however, it should be noted that such directions likewise require explicit compositional optimization.

## 6 Conclusion

In this study, we introduced an explicit optimization method for single-layer feed-forward neural networks, which moreover, demonstrated significant promise for generalized extension into multilayer networks. Our method, underpinned by insights from mathematical analyses, demonstrates near-optimal performance, with iterative optimization offering only finer enhancements, and randomly initialized parameters gradually converging towards the performance levels of explicit solutions. This includes for iterative applications of explicit solution local optimizations in multi-layer networks. We see our work as serving as a keystone, enhancing training efficiency and model interpretability, while providing insight into model function. The efficacy of our solution is substantiated through language modeling and digit classification tasks, underscoring its wide-ranging applicability and generalization potential. We anticipate that this work will catalyze further research into the application of explicit optimization methods to more intricate, compositional multi-layer architectures and attention-based models, enriching the field with new perspectives and methods.

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

## A  PROOF OF EXPLICIT FEED-FORWARD OPTIMIZATION

**Theorem**: A softmax-activated feed-forward layer receiving $K$-norm non-negative $D$-dimensional inputs $\boldsymbol{H}_{m,:}$ for each target of prediction $\boldsymbol{Y}_{m,:}$ is approximately optimized by a column-wise translation of the layer's generalized log-co-occurrence matrix: $\boldsymbol{U}_{j,i} = \log \boldsymbol{F}(\boldsymbol{H},\boldsymbol{Y})_{j,i} + \log \boldsymbol{w}_i$. The translating weights, $\log \boldsymbol{w}_i$, are defined by $i$-column (output) as: $\log \boldsymbol{w}_i = \frac{K-1}{K} \log(\sum_{d=1}^{D} \boldsymbol{F}(\boldsymbol{H},\boldsymbol{Y})_{d,i})$, defining an explicit form for each of the layer's $j,i$-parameters by the expression:

$$\boldsymbol{U}_{j,i} = \log \boldsymbol{F}(\boldsymbol{H},\boldsymbol{Y})_{j,i} - \frac{K-1}{K} \log \left( \sum_{d=1}^{D} \boldsymbol{F}(\boldsymbol{H},\boldsymbol{Y})_{d,i} \right) \tag{5}$$

**Proof**: Abbreviating $\boldsymbol{F}(\boldsymbol{H},\boldsymbol{Y})$ by simply $\boldsymbol{F}$ for concise notation, first re-arrange the starting expression for a $j,i$-index pair of $\boldsymbol{U}$ is:

$$\boldsymbol{U}_{j,i} = \log \boldsymbol{w}_i \boldsymbol{F}_{j,i} \tag{6}$$

It is a matter of algebra to reduce the the likelihood function's general form with $\boldsymbol{w}$ to an expression only dependent on $\boldsymbol{w}$'s components in the denominator-factors:

$$e^L = \prod_{m=1}^{M} \frac{e^{\boldsymbol{H}_{m,:}\boldsymbol{U}_{:,\boldsymbol{i}_m}}}{\sum_{n=1}^{N} e^{\boldsymbol{H}_{m,:}\boldsymbol{U}_{:,n}}} = \prod_{m=1}^{M} \frac{\prod_{d=1}^{D} \boldsymbol{F}_{d,\boldsymbol{i}_m}^{\boldsymbol{H}_{m,d}}}{\boldsymbol{w}_{\boldsymbol{i}_m}^{-K} \sum_{n=1}^{N} \boldsymbol{w}_n^K \prod_{d=1}^{D} \boldsymbol{F}_{d,n}^{\boldsymbol{H}_{m,d}}} \tag{7}$$

Above, the expression shows that one need only minimize the denominator at right to maximize the overall expression, i.e., optimize the likelihood. Since the logarithm is a monotone function, this is likewise equivalent to maximizing the logarithm of the denominator, which we denote by $\Upsilon$:

$$\Upsilon = \sum_{m=1}^{M} \epsilon_m = \sum_{m=1}^{M} \log \left[ \boldsymbol{w}_{\boldsymbol{i}_m}^{-K} \sum_{n=1}^{N} \boldsymbol{w}_n^K \prod_{d=1}^{D} \boldsymbol{F}_{d,n}^{\boldsymbol{H}_{m,d}} \right] \tag{8}$$

We then proceed directly, by differentially optimizing $\Upsilon$ and compute partial derivatives of $\epsilon_m$:

$$\frac{\partial \epsilon_m}{\partial \boldsymbol{w}_i}\bigg|_{\boldsymbol{i}_m=i} = \frac{K \boldsymbol{w}_i^{K-1} \prod_{d=1}^{D} \boldsymbol{F}_{d,i}^{\boldsymbol{H}_{m,d}}}{\sum_{n=1}^{N} \boldsymbol{w}_n^K \prod_{d=1}^{D} \boldsymbol{F}_{d,n}^{\boldsymbol{H}_{m,d}}} - \frac{K}{\boldsymbol{w}_i} \quad ; \quad \frac{\partial \epsilon_m}{\partial \boldsymbol{w}_i}\bigg|_{\boldsymbol{i}_m \neq i} = \frac{K \boldsymbol{w}_i^{K-1} \prod_{d=1}^{D} \boldsymbol{F}_{d,i}^{\boldsymbol{H}_{m,d}}}{\sum_{n=1}^{N} \boldsymbol{w}_n \prod_{d=1}^{D} \boldsymbol{F}_{d,n}^{\boldsymbol{H}_{m,d}}} \tag{9}$$

Putting these pieces together in-sum produces the expression:

$$\frac{\partial \Upsilon}{\partial \boldsymbol{w}_i} = -\frac{K \boldsymbol{f}_i}{\boldsymbol{w}_i} + \sum_{m=1}^{M} \frac{K \boldsymbol{w}_i^{K-1} \prod_{d=1}^{D} \boldsymbol{F}_{d,i}^{\boldsymbol{H}_{m,d}}}{\sum_{n=1}^{N} \boldsymbol{w}_n \prod_{d=1}^{D} \boldsymbol{F}_{d,n}^{\boldsymbol{H}_{m,d}}} \tag{10}$$

it becomes helpful now to identify a weighted, geometric mean of co-occurrences with token $i$ over the $m^{\text{th}}$ instance's features: $\mathbb{E}_G[\boldsymbol{F}_{j,i} \mid \boldsymbol{H}_{m,:}] = (\prod_{d=1}^{D} F_{d,i}^{\boldsymbol{H}_{m,d}})^{1/K}$. Provided their inner product with the weights is approximately a constant $c \in \mathbb{R}$:

$$\sum_{n=1}^{N} \boldsymbol{w}_n \mathbb{E}_G[\boldsymbol{F}_{j,n} \mid \boldsymbol{H}_{m,:}] \approx c, \tag{11}$$

solving for $\frac{\partial \Upsilon}{\partial \boldsymbol{w}_i} = 0$ results in the following proportionality for each token-index, $i$:

$$\frac{\boldsymbol{f}_i}{\boldsymbol{w}_i^K} = \sum_{m=1}^{M} \frac{\mathbb{E}_G[\boldsymbol{F}_{j,i} \mid \boldsymbol{H}_{m,:}]^K}{\sum_{n=1}^{N} \boldsymbol{w}_n \mathbb{E}_G[\boldsymbol{F}_{j,n} \mid \boldsymbol{H}_{m,:}]^K} \propto \sum_{m=1}^{M} \mathbb{E}_G[\boldsymbol{F}_{j,i} \mid \boldsymbol{H}_{m,:}]^K \tag{12}$$

Each $\mathbb{E}_G[\boldsymbol{F}_{j,i} \mid \boldsymbol{H}_{m,:}]$ likely correlates to $\boldsymbol{f}_i$, and their sum further integrates a broader average:

$$\sum_{m=1}^{M} \mathbb{E}_G[\boldsymbol{F}_{j,i} \mid \boldsymbol{H}_{m,:}]^K = M \langle \mathbb{E}_G[\boldsymbol{F}_{j,i} \mid \boldsymbol{H}_{m,:}] \rangle^K \tag{13}$$

Here, the expression $\langle \mathbb{E}_G[\boldsymbol{F}_{j,i} \mid \boldsymbol{H}_{m,:}] \rangle$ indicates the $K$-power mean *of* the geometric means of co-occurrences with token $i$. We thus find that an explicit form for $\boldsymbol{w}_i$'s proportionality is:

$$\boldsymbol{w}_i \propto \frac{\boldsymbol{f}_i^{1/K}}{\langle \mathbb{E}_G[\boldsymbol{F}_{j,i} \mid \boldsymbol{H}_{m,:}] \rangle} \propto \boldsymbol{f}_i^{\frac{1-K}{K}} = \left( \sum_{d=1}^{D} \boldsymbol{F}(\boldsymbol{H},\boldsymbol{Y})_{d,i} \right)^{\frac{1-K}{K}}, \tag{14}$$

dependent on the double-averaged denominators scaling with count: $\langle \mathbb{E}_G[\boldsymbol{F}_{j,i} \mid \boldsymbol{H}_{m,:}] \rangle \propto \boldsymbol{f}_i$. ∎

## B  ADDITIONAL LOSS CURVES

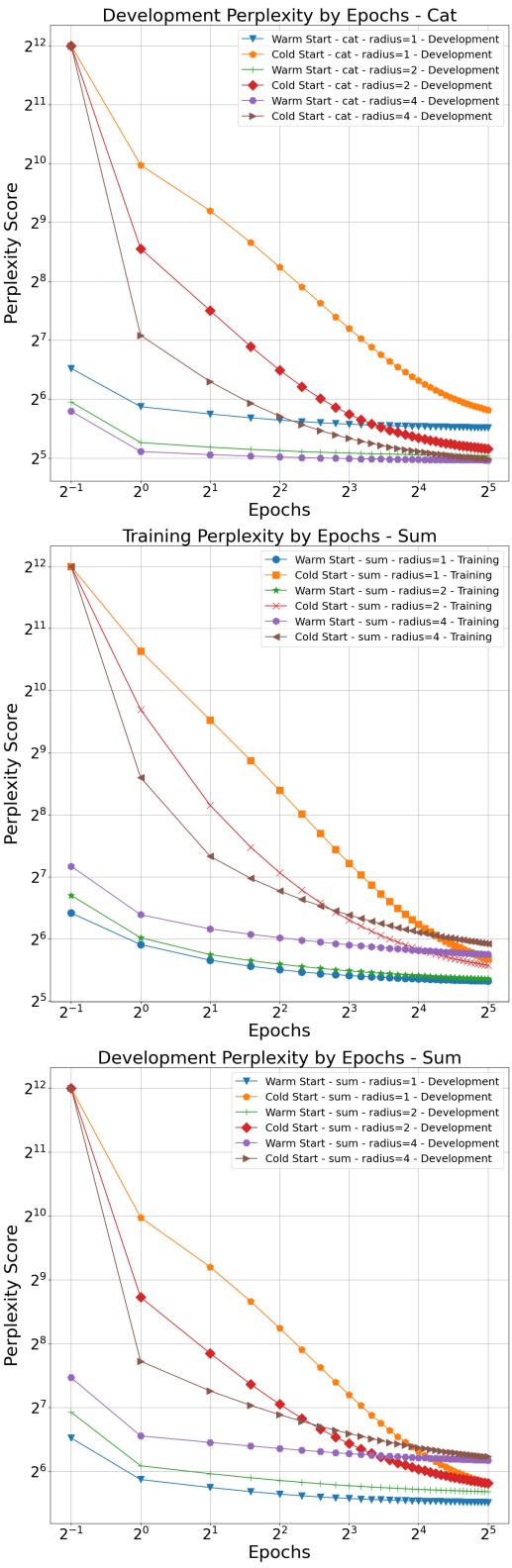

Figure 3: Development set loss curves from iterative optimization in the case of cold (random parameters) and warm (explicit solution parameters) start for the **Cat** model, alongside similar plots for both training and development sets and the **Sum** model. Axes depict the perplexity as a function of epoch (iteration) on a logarithmic scale for a total of 32 epochs. Note: model loss *prior* to training is likewise reported in the loss curves, with values presented at epoch $0.5$ for visual clarity, as a means of observing the degree of optimally achieved by the explicit solution as a starting point.

