# OpenReview forum: "Reducing the Need for Backpropagation and Discovering Better Optima With Explicit Optimizations of Neural Networks"
_ICLR.cc/2024/Conference — ICLR 2024 Conference Withdrawn Submission_

### Official Review · Reviewer_7Z9Y · 2023-10-13

**Soundness:** 2 fair
**Presentation:** 2 fair
**Contribution:** 1 poor
**Rating:** 3
**Confidence:** 3

**Summary:**

The authors propose an approximate closed-form solution for a one-layer softmax activated feed-forward neural network. They demonstrate that when warm-starting the weights of a one- and two-layer neural network, the network is already close to the optimum and requires fewer gradient-descent iterations, alleviating the need for backpropagation.

**Strengths:**

- The paper proposes an approximate closed-form solution to a one-layer softmax layer.
- The paper demonstrates that setting the weights to the approximate optimum gives you good performance which can be further improved via gradient descent.
- This saves gradient-descent iterations.

**Weaknesses:**

- I think the paper overall could be condensed much more to clearly expose the main messages. Especially, section 5 for discusssion could be made much more concise.
- As the parameter values are initialized randomly for the cold-start models, (bootstrap) confidence intervals should be reported over an appropriate number of random initializations. Similarly for the warm-start models during training.
- Figure 4 is referenced, but as far as I can tell Figure 2 is meant.
- Why is the performance of the two-layer neural network not better than the one-layer neural network for warm starting?
- No code to reproduce the experiments is provided.

**Questions:**

- There are explicit and implicit integration schemes for differential equations but explicit optimization is not clearly defined. For example, the authors could consider calling it an approximate optimization solution in closed-form.

### Typos:
These are minor and simply for completeness.
- "understanding explicit forms for **a** given softmax classifier’s inputs and outputs."
- "This separate-intercept implementation was actually
the in**i**tial (equivalent) formulation used for MNIST experiments
- Why did you report accuracy with and without % in Figure 2?

---

### Official Review · Reviewer_cCEC · 2023-10-19

**Soundness:** 3 good
**Presentation:** 2 fair
**Contribution:** 1 poor
**Rating:** 3
**Confidence:** 3

**Summary:**

In this paper, the authors present an innovative and computationally efficient approach to optimizing neural networks. They propose a method that not only reduces the computational costs of scaling neural networks but also offers highly efficient optimizations for resource-constrained applications. The approach involves deriving an explicit solution for a simple feed-forward language model by analyzing the continuous bag-of-words (CBOW) form of word2vec. They show that this solution generalizes to a wide range of single-layer feed-forward softmax-activated neural models trained on positive-valued features. The authors support their claims through computational experiments, demonstrating that their solution is near-optimal and that iterative optimization only marginally improves upon it. Additionally, they discuss the advantages of applying this solution to multi-layer networks, showing that better optima can be achieved compared to backpropagation alone, especially after a multi-layer warm-start is applied. This paper provides a promising alternative for more efficient neural network optimization, with the potential for significant computational savings and improved performance.

**Strengths:**

1. interesting results and great potential to be applied to a broad spectrum of models
2. The very creative way of designing toy experiments, especially the LM experiment, seems exciting and intuitively motivated.

**Weaknesses:**

1. The paper claims results on LMs. Still, it limits its scope to simple self-defined architectures, which significantly limits the impactfulness/usefulness of this paper, and it's disappointing to see no modern architectures, such as self-attention, are analyzed and experimented on in this paper, despite the eye-catching title and abstract.
2. Experiments are too toy-like, which might not reflect the reality of the effectiveness of the proposed method. For instance, MNIST problem can be easily solved by zeroth-order optimization methods, where no gradient/information about the network is needed. It really doesn't say much about the ability to find optimality.
3. Need more datasets beyond BabyLM to validate the assumption: "the degree to which token counts scale with their average of geometric means of co-occurrences". How could this assumption be made based on a single dataset?




Some small issues on presentations: 1) notations are messy with many sub/superscripts, the paper will be way much more readable if those are cleaned up properly. 2) contributions are not listed clearly and written in very convoluted ways both in introduction and abstract.

**Questions:**

1. What happens when in a more realistic setting, where co-occurrence can only be estimated in a stochastic fashion? What does it say about the convergence/optimality of the method under that setting?

---

### Official Review · Reviewer_h8Kh · 2023-10-27

**Soundness:** 1 poor
**Presentation:** 1 poor
**Contribution:** 2 fair
**Rating:** 3
**Confidence:** 4

**Summary:**

The submission presents a way of initializing the weights of a single-layer, autoregressive sequence model, with a softmax non-linearity at the output.
The initialization has a closed-form formula, computed from the co-occurrence matrix of tokens, which approximates the loss minimizer.
Experiments on a 10-million token dataset from the BabyLM Challenge, and MNIST, show this initialization performs better than random initialization in terms of training speed.

**Strengths:**

Better pre-initialization of neural network weights is an interesting topic, and potentially impactful. Data-dependent initialization, even limited to the output weights, have already proved useful in the few-shot learning case for instance (Qi et al., "Low-shot learning with imprinted weights", CVPR 20218).
Such solutions in the domain of token-based sequence models could be impactful.

**Weaknesses:**

Originality
---------------
Using co-occurrences to improve neural networks has already be explored in previous work, for instance:
- Kurata et al., "Improved Neural Network-based Multi-label Classification with Better Initialization Leveraging Label Co-occurrence", NAACL-HLT 2016
- Yao et al., "Incorporating Label Co-Occurrence Into Neural Network-Based Models for Multi-Label Text Classification", IEEE Access 2019

Co-occurrences have also been part of statistical NLP for a long time. However references to these earlier ideas are not included or discussed.

Quality
----------
1. The closed-form formula provided for the initialization is presented as an "explicit solution", but it's really an approximation, and relies on additional assumptions that are not clarified until the appendix.
2. Many of the experimental settings and hyperparameters are not reported even in the appendix, and hyperparameter optimizations is not reported. Variability due to various random factors is not reported either, no confidence intervals are provided.
3. No baseline for the experiments are provided for comparison. For instance, n-gram based sequence models would be a natural basis for comparisons with the simple neural-based models being explored, with the length n of the n-gram mapping naturally to the "radius" or window size K.
4. The experiments on MNIST have major clarity issues (see next section), but also their results are much worse than what logistic regression achieves on that dataset (> 90%).
5. The "scaling relationship" between average of co-occurrences average and token counts is not really made clearer by Fig. 1. I'm not sure what is depicted there or what it proves. The overlap of many points does not allow for a clear picture of the relationship. It's also not clear what would be expected in different cases.
6. The theorem seems to assume the form of a solution (Eq. 6) and proceeds to fine an approximation of `w`, but there is no proof that this gives a reasonable approximation of H, or for which error measure.

Clarity
----------
The paper was overall hard to read through, concepts were not introduced clearly, and notation was sometimes confusing.
In particular:
1. I was confused by the dimensions of various tensors. Is N the vocabulary size (number of tokens)? Is M the length of a given sequence, or the total number of examples? What's the dimension of the embeddings? Maybe a figure can help.
2. For the MNIST experiments, I did not understand how the sequential setting was adapted. In particular, the inputs and outputs do not seem to belong to the same space. Section 3.2 suggests all the images belong to the same sequence somehow. The "priming number" is never clearly defined, is the prediction done on the basis of a single pixel at a given position?
3. "Sum" vs. "Cat" aggregations are being referred to before being defined.
4. The caption for table 1 should (at least briefly) explain the results displayed.

Significance
------------------
The results are not particularly convincing compared to simple baselines

Minor points
-----------------
1. "Fig. 4" should probably be "Fig. 2"
2. The same paper (or variants of the same paper) is sometimes cited twice (Mikolov et al. 2013, Rumelhart et al. 1986)
3. The MNIST citation seems wrong, with Cortes's name combined with LeCun's.

**Questions:**

1. How does that model fare compared to n-gram based sequence models?
2. Can you clarify how sequences are used in the MNIST case, and what the input and output features are?
3. The paper states that computing the initialization weights is equivalent to performing one epoch of SGD, as it's one pass through the data. But I would expect computing the co-occurrences to scale quadratically, can you clarify?
4. Could the "scaling relationship" between co-occurrences average and token counts simply be a consequence of how the vocabulary was constructed, in particular using byte-pair encodings?
5. What's the right sign between the 2 terms of the formula for U (the equation is not consistent with the body of the theorem)

**Details Of Ethics Concerns:**

Large parts of the submission (theorem, full paragraphs) are shared with another submission I'm reviewing ([6161](https://openreview.net/forum?id=MCQdWMs5iA)), and neither acknowledge the other one.

---

### Official Review · Reviewer_59hn · 2023-11-01

**Soundness:** 1 poor
**Presentation:** 1 poor
**Contribution:** 1 poor
**Rating:** 1
**Confidence:** 4

**Summary:**

The paper proposes an initialization technique for a single layer neural network assuming positive inputs and softmax output. The initialization, which authors consider to be "near optimal" is based on simplifying assumption when analytically solving the softmax objective. The authors run several experiments on MNIST and BabyLM datasets, and show that once model initialized using proposed method it gets better perplexity/accuracy (for the same number of following SGD iterations) compared to random initialized model.

**Strengths:**

None

**Weaknesses:**

This paper has many weaknesses that must be addressed before it can be recommended for publication. Here are the major ones:
1. Presentation. The paper uses peculiar style of presentation when terms and results are used way before they are formally introduced (if at all) and often use non-standard terminology. For example, what is radius in section 2.2? What should reader conclude from fig 1 when reading same section 2.2?, How is raidal-concatenation defined? (section 3.1) What is parameter disorientation? What is cone-shaped linearity? What does it mean to *stabilize* scaling relationship?  (page 7). What is co-optimization (same page 7)? These are only some of the many questions I had when reading the paper.
2. Significance. Throughout the paper, it is claimed that proposed method of "explicit" optimization (btw, are other initializations not explicit?) can significantly accelerate the training. But so does the other initialization techniques: many initializations in NN world were proposed and some are being used everyday (e.g., Kaiming He's init, Xavier Glorot's init, and many many others). Why there is no comparison to those? Also, for a simple problem consisting of a single layer + softmax one can come-up with many others (obtain least squares fit and init from it, run k-means and start from it, etc, etc)... Therefore, it is impossible to consider the merits and significance of the proposed method without running proper comparison against other initialization techniques.
3. Limitations. However good is the proposed initialization, it only works for a single layer models with multi-layer setting being heavily discussed and theorized, but without any actual results. This is a severe limitation: I can't think of any use case that can significantly benefit from the proposed method.
4. Results. A "non trivial" 82% accuracy on MNIST with given initialization (no training) is *trivial result* in the opinion of the reviewer. A simple prototype based classifier based on distance to the mean of each digit gives accuracy of 92%+, and logistic regression would give 97%...

**Questions:**

see weaknesses